# The Isocaloric Substitution of Plant-Based and Animal-Based Protein in Relation to Aging-Related Health Outcomes: A Systematic Review

**DOI:** 10.3390/nu14020272

**Published:** 2022-01-09

**Authors:** Jiali Zheng, Tianren Zhu, Guanghuan Yang, Longgang Zhao, Fangyu Li, Yong-Moon Park, Fred K. Tabung, Susan E. Steck, Xiaoguang Li, Hui Wang

**Affiliations:** 1School of Public Health, Shanghai Jiao Tong University School of Medicine, Shanghai 200025, China; jzheng@shsmu.edu.cn (J.Z.); marcus_zhu@sjtu.edu.cn (T.Z.); yangguanghuan1995@sjtu.edu.cn (G.Y.); 2Department of Epidemiology and Biostatistics, Arnold School of Public Health, University of South Carolina, Columbia, SC 29208, USA; LZ7@email.sc.edu (L.Z.); stecks@mailbox.sc.edu (S.E.S.); 3Department of Epidemiology, Division of Cancer Prevention and Population Sciences, The University of Texas MD Anderson Cancer Center, Houston, TX 77030, USA; Fangyu.Li@uth.tmc.edu; 4Department of Epidemiology, Fay W. Boozman College of Public Health, University of Arkansas for Medical Sciences, Little Rock, AR 72205, USA; YPark@uams.edu; 5Department of Internal Medicine, Division of Medical Oncology, The Ohio State University College of Medicine and Comprehensive Cancer Center, Columbus, OH 43210, USA; Fred.Tabung@osumc.edu; 6State Key Laboratory of Oncogenes and Related Genes, Center for Single-Cell Omics, School of Public Health, Shanghai Jiao Tong University School of Medicine, Shanghai 200025, China

**Keywords:** isocaloric substitution, animal protein, plant protein, aging, health outcomes, systematic review

## Abstract

Plant-based and animal-based protein intake have differential effects on various aging-related health outcomes, but less is known about the health effect of isocaloric substitution of plant-based and animal-based protein. This systematic review summarized current evidence of the isocaloric substitutional effect of plant-based and animal-based protein on aging-related health outcomes. PubMed and Embase databases were searched for epidemiologic observational studies published in English up to 15 March 2021. Studies that included adults ≥18 years old; use of a nutritional substitution model to define isocaloric substitution of plant protein and animal protein; health outcomes covering mortality, aging-related diseases or indices; and reported association estimates with corresponding 95% confidence intervals were included. Nine cohort studies and 3 cross-sectional studies were identified, with a total of 1,450,178 subjects included in this review. Consistent and significant inverse association of substituting plant protein for various animal proteins on all-cause mortality was observed among 4 out of 5 studies with relative risks (RRs) from 0.54 to 0.95 and on cardiovascular disease (CVD) mortality among all 4 studies with RRs from 0.58 to 0.91. Among specific animal proteins, the strongest inverse association on all-cause and CVD mortality was identified when substituting plant protein for red and/or processed meat protein, with the effect mainly limited to bread, cereal, and pasta protein when replacing red meat protein. Isocaloric substitution of plant-based protein for animal-based protein might prevent all-cause and CVD-specific mortality. More studies are needed on this topic, particularly for cancer incidence and other specific aging-related diseases.

## 1. Introduction

People worldwide are living longer, and the pace of aging is increasing dramatically. According to the World Health Organization, the proportion of the world’s population over 60 years will nearly double from 12% in 2015 to 22% in 2050, with an approximate 2 billion people aged 60 and above and 434 million people aged above 80 years by 2050 [1]. During aging, numerous physiological changes contribute to decreases in physical and mental capacity, immunity, and metabolism, which altogether lead to higher cardiometabolic risk and further increase the risk of aging-associated diseases such as cancer, cardiovascular disease (CVD), type-2 diabetes (T2D), chronic kidney disease, neurodegenerative disease, sarcopenia, frailty, and ultimately death [2,3,4,5,6,7].

As an essential macronutrient for humans, dietary protein plays a key role in maintaining physiological functioning and optimizing health span and longevity, mainly because of its favorable effect on weight management, strength, and maintaining cardiometabolic profile [8,9,10,11]. In regard to the health effect of protein intake, substantial evidence has suggested that the source of protein and not only the amount of protein is important. Previous reviews reached the conclusion that animal-based and plant-based protein had differential health effects: higher consumption of animal-based protein and major contributing food sources such as red and/or processed meat were associated with increased risk of cardiometabolic markers [12], weight gain [13], CVD [14,15,16,17], kidney diseases [18], gastrointestinal cancers [19], T2D [20,21], and all-cause mortality [22], while higher intake of plant protein was linked to decreased risk of several aging-related outcomes including metabolic syndrome and its risk factors, T2D, cancer, CVD mortality, and all-cause mortality [17,20,23,24,25,26,27].

Individuals tend to keep daily calorie intake constant under normal health status; thus, it is important to acknowledge that the effect of an increased intake of a macronutrient on a given health outcome is countered by reduced energy intake of another macronutrient (i.e., a substitution effect). Theoretically, this isocaloric substitutional effect can be statistically estimated in the nutritional substitution model while leaving out the replaced macronutrient but including other macronutrients and total energy in the model [28]. Substitution analysis for plant- and animal-based protein may thus have an advantage over studies investigating the independent effect of protein intake as the substitution analysis may differentiate the health effects of these two major protein sources, and further help to identify and inform the optimal protein source for health promotion and disease prevention.

Recently, several studies were published on the topic of isocaloric substitution of animal- and plant-based protein on a wide array of aging-related health outcomes [7,10,29,30,31,32,33,34,35,36,37,38]. However, summary evidence on this topic is lacking. Therefore, we conducted this review to add new evidence in the literature by comprehensively summarizing the current evidence of the isocaloric substitutional effects of animal- and plant-based protein on various aging-related outcomes.

## 2. Materials and Methods

This systematic review was conducted and reported according to the PRISMA (Preferred Reporting Items for Systematic Reviews and Meta-Analyses) guidelines [39]. This review was registered with the International Prospective Register of Systematic Reviews (PROSPERO identifier: CRD 42021258075).

### 2.1. Search Strategy

A comprehensive literature search of epidemiological observational studies on the association between isocaloric substitution of plant-based and animal-based protein and aging-related health outcomes was conducted using the PubMed and Embase databases. Each search term related to dietary plant or animal protein intake (“protein intake”, “dietary protein”, “animal-based protein”, “animal protein”, “plant-based protein”, “plant protein”, “protein source”, “food source”, “food”, “diet”) in combination with each term associated with isocaloric substitution (“substitution”, “substitute”, “replace”, “replacement”, “alternative”, “isocaloric”, “isoenergetic”) was searched using an “AND” command in two databases. The search was restricted to human studies published up to 15 March 2021 in English. Due to the broad scope of aging-related diseases, we did not restrict outcomes in the search phase.

### 2.2. Eligibility Criteria

Studies that met the following PICOS (Population, Intervention, Comparison, Outcomes, Study design) criteria were included in this systematic review (Table 1): (1) cross-sectional, case-control, or cohort study as study design; (2) studies were conducted in the adult population aged over 18 years; (3) the nutritional substitution model with the replaced protein left out of the model but the other macronutrients and total energy included was clearly demonstrated; (4) isocaloric substitution of plant-based protein intake (or plant-based protein from specific food sources) for animal-based protein intake (or animal-based protein from specific food sources) was described as the exposure; (5) aging-related health outcomes including all-cause and cause-specific mortality, cancers, CVD, cerebral diseases, cognitive impairment, chronic kidney disease, respiratory diseases, risk markers of metabolic diseases and aging-related indices were described as study outcomes; and (6) studies provided odds ratios (ORs), hazard ratios (HRs), prevalence ratios (PRs) or relative risks (RRs) with corresponding standard errors or 95% confidence intervals (CIs). Clinical trials were excluded in this review because the substitutional effects were observed in an interventional approach in clinical trials while it was generated from an ideal and theoretical model-based approach in observational studies. If the substitutional associations were obtained only based on combined data from pooling cohorts or results from meta-analysis of several separate cohorts without estimates from individual studies, we treated this as one study. References from identified publications or relevant reviews were also searched to identify additional eligible studies.

### 2.3. Study Selection and Data Extraction

Two authors (T.Z. and G.Y.) independently searched the literature and selected the eligible studies based on prespecified inclusion criteria. The study selection process is shown in Figure 1. The following important characteristics from the included studies were extracted into Table 2: reference of the publication (name of the first author and year of publication); study design and study location; number of cases and total participants at risk; mean age or age range of baseline study population; mean or median follow-up time for cohort studies; characteristics of dietary exposure; the substitution model; study outcome; and variables from the most fully-adjusted model. We described the substitutional effect of animal protein for plant protein or plant protein for animal protein separately and specified the food sources of protein for studies describing replacement of protein from food sources. For the substitution model, we described the type (continuous or categorical) and unit (e.g., percentage of total energy) of the substituted exposure and the other macronutrients and total energy that were held constant in the model. Forest plots were used to show the point estimates (e.g., RR, PR, HR, OR) and 95% CIs of each reported substitutional association of animal-based and plant-based protein with aging-related disease mortality risk or associations with disease incidence/prevalence risk based on the findings from the most fully adjusted model, along with the *P*-trend value for the highest versus lowest category of the substituted exposure if available. Sex-specific associations were also displayed if they were presented in the original studies. In each forest plot, associations were listed by outcomes from an overall disease to its subtypes (e.g., CVD overall and specific CVD disease such as stroke). For the same outcome, exposures were arranged from substitution of total plant protein for total animal protein and next, substitution of total plant protein for specific animal proteins, followed by specific plant proteins replacing specific animal proteins. A meta-analysis was not conducted in the current review because of the small number of studies on the exact same exposure of substitution (i.e., same specific animal and plant protein for substitution) and the same aging-related outcome. All forest plots were drawn using R software (R 3.6.3).

### 2.4. Quality Assessment

The methodological quality of included studies was also independently assessed by two authors with reference to the Newcastle-Ottawa Quality Assessment Scale (NOS) [40]. Stars were assigned to each study based on NOS criteria for study selection, comparability and outcome, with a maximum of 9 stars for cohort study and 10 stars for cross-sectional studies, and studies with 7 or more stars were considered to be of adequate quality [40]. The discrepancies in the study selection and quality assessment were resolved by consulting with a third reviewer (J.Z.).

## 3. Results

A total of 1822 records were identified from PubMed and Embase after removing duplicates, among which full-text review was conducted for 103 articles. After excluding studies not meeting the inclusion criteria with reasons listed in Figure 1, 12 articles published between 2005 and 2021 were included for final analyses [7,10,29,30,31,32,33,34,35,36,37,38]. Among these, 1 article described substitution results only based on the combined dataset of 2 prospective cohorts [10] and another reported substitutional effects based on fixed-effects meta-analysis of 3 prospective cohorts without individual study results [33], while all the others described only 1 population in the study; therefore, we reported associations from a total of 12 studies in the present review. In the assessment of the study quality, all 12 studies scored 7 and above according to NOS, so they were all retained in this review.

### 3.1. Study Characteristics

We included 9 prospective cohort studies [7,10,29,30,31,32,33,35,37] and 3 cross-sectional studies [34,36,38] with a total of 1,450,178 participants aged from 15 to 79 years, in the review (Table 2). Two cross-sectional studies were conducted at baseline within respective European cohort studies [34,38], and another cross-sectional analysis was conducted within a clinical trial, i.e., European Diet, Obesity and Genes Trial (DIOGenes) [36]. Of the 9 cohort studies, 6 were conducted in the United States [10,30,31,32,33,35], 2 in Europe [7,37] and 1 in Japan [29], with mean or median follow-up time ranging from 7 years [37] to the longest 27 years [10]. A total of 156,064 overall deaths including 55,505 deaths from cancer and 41,836 deaths from CVD were identified from 5 cohort studies [10,29,30,31,35] while a total of 29,851 incident cases of various aging-related diseases were identified from the other 4 cohort studies [31,32,33,37], which consisted of 13,838 cancer cases, 15,580 T2D cases, 298 hypertension cases, and 135 microalbuminuria cases. Two studies in the review included female participants only [31,35], and one study reported sex-specific associations only [30], whereas all the other studies presented substitutional associations among men and women combined. Other than two studies conducted among diseased populations (i.e., type-1 diabetes [37] and T2D [38]), the other studies in this review were conducted in disease-free populations.

The food frequency questionnaire (FFQ) was the most common instrument used to assess diet intake (n = 8) [10,29,30,31,32,33,35,38] with 122 to 177 included food items and was self-administered in all of the 8 studies. Food records were applied in 3 studies for participants to carefully document either a 3-day [36,37] or 1-week diet [34]. In the other study, a computerized diet history covering 900 items was used [7]. Diet was measured only once in the majority of studies; however, in two Harvard pooling cohort studies diet was measured repeatedly at baseline and every 4 years thereafter, and cumulative average dietary intake was analyzed [10,33] and diet was measured at 2 time points in 2 studies to calculate the average [36] or change [7] of protein intake between the 2 measurements respectively. FFQ or the computerized diet history evaluated diet over the previous 1 year in the majority of studies (n = 7), over the past 3 months in one study [35], and in the past month in another study [38]. Both validity and reproducibility of diet questionnaires were assessed in seven studies [10,29,30,31,32,33,35], and only validity was assessed in two studies [7,38].

In regard to the substituted protein exposure in the isocaloric substitution model, nine used “percentage of energy” as the unit of the substituted exposure [7,10,29,30,31,33,35,37,38] while the other three studies used percentage of total protein [36], quintiles of amount of protein intake [32], and grams of protein per body weight [34]. Among the 9 studies using “percentage of energy” as protein substitution unit, 5 reported the substitutional associations with the unit as 3% of energy from animal-based or plant-based proteins for substitution [10,29,30,37,38], 2 reported the protein substitutional effect based on each 5% of energy intake [33,35], 1 on each 1% of energy intake [7], and another one comparing the highest versus lowest quintile of the percentage of energy from plant proteins substituted for animal proteins [31]. In two studies [37,38], both isocaloric substitutional effects of plant protein for animal protein and animal protein for plant protein were reported, while other studies only reported the effect of plant protein intake substituted for animal protein intake. While seven studies only reported the substitutional effect of total plant and total animal protein [31,33,34,35,36,37,38], the other five studies included investigation on protein from specific food sources [7,10,29,30,32]. In a majority of substitution models, the isocaloric substitution of plant and animal protein was calculated by including all macronutrients except the replaced exposure (n = 5) or by controlling for total protein intake (n = 3), but carbohydrate was accounted for by including vegetables, fruits, glycemic index or fiber instead of controlling for carbohydrate directly in the other studies [10,30,33,35]. The aging-related health outcomes reported in this review included all-cause and cause-specific mortality outcomes [10,29,30,31,35], T2D [33], total cancer and colorectal cancer [31,32], hypertension [37], renal function impairment and microalbuminuria [37,38], cardiometabolic risk markers [36] and two aging-related indices, i.e., sarcopenia risk score (SRS) [34] and deficit accumulation index (DAI) [7]. In the multivariable models, most studies adjusted for important confounders associated with dietary protein intake and disease, including age, sex, body mass index, physical activity, alcohol consumption, cigarette smoking, personal medical history, total energy intake and other macronutrients (n = 8); of these, three additionally adjusted for family history of the relevant disease [31,32,35] and four additionally adjusted for socioeconomic status [7,30,32,35].

### 3.2. Substitution of Protein from Animal and Plant Sources and Mortality Outcomes

Five cohort studies (four in the United States [10,30,31,35] and one in Japan [29]) reported the associations of various plant-based proteins substituted for different types of animal-based proteins with mortality outcomes including all-cause mortality, CVD-specific mortality, total cancer mortality, and respiratory disease mortality (Appendix A) [10,29,30,31,35].

Except for the Iowa Women’s Health Study (IWHS) in which quintiles of total plant protein intake were analyzed as the substituted exposure [35], all 5 studies on all-cause mortality consistently supported a significant lower all-cause mortality risk when substituting 3% or 5% energy intake from total plant protein for total animal protein [30,35] and specific animal proteins including red and/or processed meat [10,29,30], white meat [30], poultry [10], fish [10,29], and eggs [10,29,30] with RRs ranging from 0.54 to 0.95 (Appendix A). Among these, the strongest risk reduction was observed for substituting 3% energy from total plant protein for processed meat protein (HR = 0.54, 95% CI = 0.38–0.75), which was identified in the Japan Public Health Center-based (JPHC) cohort [29] with 12,381 total deaths after a mean of 18 years of follow-up, followed by substitution for red meat protein (HR = 0.66, 95% CI = 0.55–0.80) in the same JPHC cohort and for processed red meat (HR = 0.66, 95% CI = 0.59–0.75) based on the finding from the pooling cohort of the Nurses’ Health Study (NHS) and the Health Professionals Follow-up Study (HPFS) [10]. However, inverse association between substitution of plant protein for dairy protein with all-cause mortality was only seen in two U.S. cohorts [10,30] but not in the Japanese cohort [29]. In the U.S. National Institutes of Health–American Association of Retired Persons (NIH–AARP) study by Huang et al., substituting specific plant proteins in addition to total plant protein for red meat protein and egg protein in relation to all-cause mortality was also examined [30]. For both males and females, among different plant proteins, the significant reduced all-cause mortality was limited to bread, cereal and pasta protein when replacing red meat protein. Changing from egg protein to all-plant proteins except nut protein significantly reduced all-cause mortality [30].

In terms of total cancer mortality, four prospective cohort studies reported its association with isocaloric substitution of plant protein for animal protein (Appendix A) [10,29,30,31]. All the studies used 3% energy intake from plant protein substituted for animal protein as the exposure except in the IWHS study, in which substitution of highest vs lowest quintile of percentage of energy from plant protein for animal protein was analyzed [31]. A significant lower cancer mortality risk was demonstrated for increased intake of total plant protein to the detriment of total animal protein among females but not males in the NIH–AARP cohort [30], while there was an association in the IWHS cohort where a different type of substitution exposure was applied [31]. When examining substitutional effects for specific sources of animal protein, a change from red meat protein to plant protein was significantly associated with a 7% to 39% reduced cancer mortality risk in the JPHC and NIH–AARP cohorts [29,30], but in the pooled NHS and HPFS, neither processed red meat protein nor unprocessed red meat protein was associated with cancer mortality when replaced with plant protein [10]. In general, substituting plant protein for overall white meat protein or specific white meat proteins (i.e., poultry and fish protein) as well as dairy protein was not associated with cancer mortality, and the substitutional effect of plant protein for egg protein was not consistent [10,29,30]. Moreover, in the analysis of specific plant protein sources in the NIH–AARP study, substitution of 3% energy from plant protein from bread, cereal and pasta for either red meat protein or egg protein significantly lowered cancer mortality risk by 12% in men and 13% in women, and by 18% in men and 20% in women, respectively [30]. Beans and legumes protein significantly reduced cancer mortality among women only when substituted for red meat or egg protein. Other plant proteins were not associated with cancer mortality risk among both sexes when replacing red meat or egg protein [30].

Four cohort studies (three studies in the United States [10,30,35] and one in Japan [29]) reported associations between CVD-specific mortality and substitution of plant proteins for animal proteins (Appendix A). Significantly lower CVD mortality risk was associated with substitution of total plant protein for total animal protein in both sexes, with HRs ranging from 0.78 to 0.89 [30,35], and the significant lower risk was also confirmed when total plant protein replaced various specific animal proteins, including total red meat, processed red meat and unprocessed red meat, white meat, and poultry with HRs varying from 0.58 to 0.91 [10,29,30]. When examining specific sources of plant protein in replacing 3% energy from egg protein and red meat protein, reduced risk of CVD mortality was only observed with substitution of bread, cereal and pasta protein for red meat protein (HRs for both sexes = 0.76) and egg protein (HR = 0.67 for males and HR = 0.69 for females) in the NIH–AARP study, plant protein from other sources (such as soy protein) was also related to reduced CVD mortality when replacing the same amount of egg or red meat protein but was only found among females [30]. When it comes to mortality from specific CVD diseases, the mortality risk of coronary heart disease was significantly reduced by 30% (HR = 0.70, 95% CI = 0.51–0.98) when comparing an average substitution of 6.1% versus 3.7% energy from total plant protein for total animal protein in the Iowa menopausal women cohort (Appendix A) [31]. The associations with heart disease-specific and stroke-specific mortality were only investigated in the NIH–AARP study where replacement of total animal protein with total plant protein significantly reduced mortality from heart disease by 10% in both sexes and the protective effect was consistently presented when exchanging plant protein for different animal proteins including red meat protein, dairy protein, and egg protein with HRs ranging from 0.72 for egg protein to 0.92 for dairy protein (Appendix A) [30]. When examining different food sources of plant protein, investigators only observed significant lower heart disease mortality when bread, cereal and pasta protein replaced red meat protein (HR = 0.77 for males and HR = 0.78 for females) or egg protein (HR = 0.68 for males and HR = 0.69 for females) in this American retired population. Similarly, as for stroke mortality, substitution of total plant protein for a variety of animal proteins from overall animal protein, red meat protein and dairy protein was all found to be significantly associated with a reduced mortality risk among both sexes, with HRs from 0.75 to 0.81 (Appendix A). The significant effect was also limited to the plant protein from bread, cereal, and pasta when replacing red meat or egg protein [30].

Findings from the NIH–AARP study revealed that significantly reduced respiratory disease mortality was only observed with substitution of 3% energy from total plant protein for red meat protein (HR = 0.83 for males and HR = 0.80 for females) and egg protein (HR = 0.61 for males and HR = 0.66 for females) (Appendix A) [30]. Bread, cereal, and pasta protein was the only source of plant protein that was associated with lower respiratory disease mortality when replacing red meat protein. Both bread, cereal, and pasta protein and protein from other plant sources reduced respiratory disease mortality when replacing egg protein. However, a significantly increased risk was reported in exchanging plant protein for white meat protein, but this only occurred in males (HR = 1.17, 95% CI = 1.03–1.33) [30]. As for dementia mortality, a significant lower mortality risk was reported in the WHI study where 5% energy of animal protein was replaced with total plant protein (HR = 0.81, 95% CI = 0.68–0.97) [35].

### 3.3. Substitution of Plant Protein for Animal Protein and Risk of Aging-Related Diseases

A total of six studies reported isocaloric substitutional effects of plant and animal protein intake on incidence or prevalence risk of different aging-related diseases: two studies focused on total cancer [31] or colorectal cancer (CRC) risk [32], one study analyzed data of three cohorts reported T2D risk [33], one on hypertension risk [37], and two studies investigated chronic kidney disease and microalbuminuria risk (Figure 2) [37,38]. No association for substitution of total plant for total animal protein on overall cancer risk was observed in the IWHS (RR = 0.99, 95% CI = 0.87–1.13) [31]. In contrast, the NIH–AARP study reported an inverse association with overall CRC risk (HR_Q5 VS. Q1_ = 0.91, 95% CI = 0.83–0.99; *P*_trend_ = 0.04), rectal cancer risk (HR_Q5 VS. Q1_ = 0.84, 95% CI = 0.71–1.00), and distal colon cancer risk (HR_Q5 VS. Q1_ = 0.84, 95% CI = 0.71–0.99) but not with proximal colon cancer risk. The inverse associations appeared to be limited to substituting total plant protein for red meat protein [32]. When further evaluating different plant proteins, the significant reduction in CRC risk was only limited to the substitution of protein from bread, cereal, and pasta for red meat protein, and this protective effect was stronger for rectal cancer and distal colon cancer than proximal colon cancer [32].

In the fixed-effect meta-analysis of the NHS, NHS II and HPFS cohorts where 205,802 participants were followed up for an average of 20 years with 15,580 incident T2D cases documented, T2D was significantly reduced by 23% (95% CI = 0.70–0.84) when substituting 5% of energy intake from total plant protein for total animal protein; consistently, a 2% to 21% reduced T2D risk was observed when substituting 1 serving per day of plant protein foods for an equal exchange of animal protein foods [33].

In the EURODIAB Prospective Complications Study with the aim to investigate baseline dietary protein intake in relation to risk of hypertension and microalbuminuria as an early clinical sign of nephropathy in 2364 type-1 diabetics from 16 European countries, no associations of hypertension and microalbuminuria risk were found with either replacement of 3% of energy from plant protein with animal protein or replacement of 3% of energy from animal protein with plant protein [37]. In the cross-sectional analyses conducted at baseline of the Dutch DIAbetes and LifEstyle Cohort Twente-1 (DIALECT-1) cohort among 420 patients with T2D, substitution of 3% energy intake from total plant protein for total animal protein significantly reduced prevalence of renal function impairment also known as chronic kidney disease (prevalence ratio = 0.20, 95% CI = 0.07–0.63) [41], but in a reverse way, replacing 3% energy from plant protein with animal protein was associated with a 31% non-significant increased prevalence risk of chronic kidney disease [38].

### 3.4. Substitution of Plant Protein for Animal Protein and Cardiometabolic Risk Markers

A cross-sectional analysis within the European DIOGenes trial was conducted in 489 overweight and obese participants. Average dietary intake was assessed with two 3-day food records for each participant in a 26-week weight maintenance intervention period when changes in body weight, body fat, and cardiometabolic risk markers were also measured [36]. No association was found for change of any cardiometabolic risk markers including body composition, serum lipids, insulin homeostasis and inflammatory biomarkers, systolic and diastolic blood pressure with increasing 1% of animal protein at the cost of plant protein.

### 3.5. Substitution of Plant Protein for Animal Protein and Indices Associated with Unhealthy Aging

Two studies reported significant inverse associations between the substitution of plant protein for animal protein and *a priori* indices associated with unhealthy aging among older adults [7,34]. In the Seniors-Study on Nutrition and Cardiovascular Risk in Spain (Seniors-ENRICA) cohort, change of DAI from baseline at 2008–2010 (wave 0) to 2017 (wave 3) was measured among 812 participants with a mean age of 68.6 years old, with higher DAI indicating higher degree of deficit accumulation that covered 4 domains of unhealthy aging, including functional impairments, self-reported vitality, mental health, and morbidities. DAI was significantly reduced by 0.46%, 0.52% and 0.47% with each 1% energy intake increase from baseline to wave 1 (2012) in the substitution of total plant protein for total animal protein, dairy protein, and meat protein, respectively [7]. At baseline of the European Project on Nutrition in Elderly People (NU-AGE) cohort of European adults aged 65–79, the SRS was measured to represent muscle quantity and strength with a higher score indicating lower muscle health. SRS was significantly reduced with each 0.1 g/body weight increase of plant protein intake to the detriment of animal protein while keeping total protein intake constant, across different levels of total protein intake (Figure 3) [34].

## 4. Discussion

Published evidence identified in this systematic review supported a significantly reduced risk of all-cause and CVD-specific mortality in relation to isocaloric substitution of total plant protein for animal proteins from various sources. The strongest protective effect was mainly observed when substituting plant protein for red and/or processed meat protein. Further examination of specific plant proteins revealed the reduced mortality risk was limited to bread, cereal, and pasta protein when replacing red meat protein. Too few studies were identified on other aging-related health outcomes such as cancer incidence, T2D, and indices associated with unhealthy aging to draw firm conclusions, though there was suggested reduced risk of incident CRC and T2D, as well as decreased unhealthy aging scores in association with isoenergetic substitution of plant protein for animal protein.

Our finding of the protective role of isocaloric substitution of plant protein for animal protein against several aging-related health outcomes had inherent consistency in many ways. Significant reduced risk of all-cause mortality, CVD mortality as well as subtypes of CVD mortality (i.e., stroke- and heart-disease-specific mortality) was consistently observed with total plant protein replaced by multiple animal proteins, including red and/or processed meat, egg, and dairy protein [10,29,30,35], which indicated exchanging plant protein for animal protein regardless of animal food sources can be protective against mortality risk, particularly CVD disease mortality. We also observed a significant protective substitutional effect of plant protein for red meat protein on all-cause mortality and CVD mortality, with the effect consistent for processed and unprocessed red meat [10,29,30], suggesting the protein substitutional effect was independent of meat processing. Consistency in substitutional associations was also reflected by the observations that the protective effect of substituting plant proteins for animal proteins existed not only for a broader or overall disease category but also for their subtypes (i.e., reduced mortality risk of CVD and CVD subtypes such as heart disease and stroke [10,29,30,35], overall CRC risk and by subsite [32]). Two aging-related indices (i.e., SRS and DAI) reflected the aging-related health status from different aspects with a higher score indicating a higher degree of unhealthy aging, and they were both significantly reduced when replacing total animal protein with total plant protein [7,34]. Huang et al. conducted sex-specific substitutional associations in the NIH–AARP study that generated similar findings across both sexes for multiple mortality outcomes, except that females tended to have a stronger protective effect from substituting other plant protein such as soy protein for red meat protein against all-cause and CVD mortality than males, which could be the differential gender effect of soy and isoflavones intake on CVD mortality that was supported in previous cohort studies [30,42,43]. In a large Japanese cohort of 40,462 participants aged between 40 to 59 years old, an inverse association between isoflavone intake and risk of myocardial infarctions and ischemic CVD mortality was observed among women only [42]. Similarly, in a cross-sectional study among 2811 Chinese men and women, habitual soy protein intake had sex-dependent effects on the risk of metabolic syndrome, a clinical risk factor for CVD; while women had lower risk associated with higher soy protein intake, men had higher risk [43]. In an intervention study of 41 hypercholesterolemic men and postmenopausal women who underwent high- and low-isoflavone soy food test phases, significantly higher interleukin-6 values after the high-isoflavone soy diet only appeared in women, indicating an estrogenic effect of soy isoflavones in enhancing the immune response among women, and thus the enhanced immune surveillance could lower risk of proinflammatory diseases such as CVD and cancer [44]. Although estrogen exerts a cardioprotective effect, there are still conflicting data regarding the beneficial or adverse effects of phytoestrogens on cardiovascular health, as phytoestrogens may act as both estrogen agonists and antagonists [45].

We observed an unexpected significant positive association between substitution of plant protein for white meat protein and respiratory disease mortality among men only in the NIH–AARP study [30], different from the substitutional associations observed with other mortality outcomes in this population. Although it was not clear which food source or component contributed primarily to white meat and plant protein in this substitutional association, the unexpected opposite direction suggested a stronger inverse association of white meat protein than plant protein with mortality from respiratory disease, or a stronger positive association of plant protein compared to white meat protein with respiratory disease mortality among males in this cohort, or this could just be a chance finding. We also noticed the different associations with total cancer mortality and all-cause mortality when using categorical (Quintile 5 vs. Quintile 1) versus continuous exposure (3% energy) of total plant protein substituted for total animal protein as the exposure, with the former reaching no association but an inverse association from the latter exposure [30,31]. This review included two studies that investigated both substitution of plant protein for animal protein and animal protein for plant protein as exposures with same outcome [37,38]; interestingly, the two-way substitution did not generate reciprocal estimates. This could be due to the two different exposure variables in the same adjusted model that often produce uncorrelated estimates statistically, and may also be due to the different strengths and significance of independent effects of animal and plant proteins on the outcome. This suggested the substitutional effect on disease outcome was not only dependent on the substitute and the one being replaced, but also dependent on the direction of the substitution.

This review only included studies applying the “leave-one-out” nutritional substitution model, with interpretation focusing on the substitutional effect of the protein in the model for the protein left out of the model with other macronutrients and total energy intake controlled as a constant. The interpretation from this model is different from the other “independent effect” nutritional model with no macronutrient left out that expresses the independent effect of a macronutrient of interest on a given outcome with adjustment for all other macronutrients. In general, independent effects of animal and vegetable proteins on all-cause and CVD mortality risk and CRC and T2D incidence risk summarized from previous literature support the plant and animal protein substitutional effects we summarized in this review. A recent large meta-analysis of 31 prospective cohort studies searched until December 2019 with 16,429 CVD deaths and 22,303 cancer deaths occurred among 715,128 participants reported that intake of total plant protein was significantly associated with a lower risk of all-cause mortality (pooled RR = 0.92) and CVD mortality (pooled RR = 0.88), but not with cancer mortality [25]. Particularly, per 90 g/day increase in whole grain intake was found to be significantly associated with 12% to 22% reduced mortality risk for CVD diseases including coronary heart disease and stroke, and intakes of specific types of whole grains including whole grain bread, total bread, and breakfast cereals were also associated with reduced risks of CVD and all-cause mortality [46]. On the contrary, there was a small positive association between unprocessed red meat intake and processed meat intake with all-cause and CVD mortality in a meta-analysis of 17 prospective cohorts [47]. In the NIH–AARP study, substitution of plant protein, particularly protein from bread, cereal, and pasta, for egg protein was associated with a strong decreased risk of stroke and heart disease mortality; however, previous literature did not consistently suggest a positive link between egg consumption and overall cardiovascular disease events or stroke, and therefore, the observed significant inverse association with substitution for egg protein possibly comes from plant protein, especially cereal protein’s cardiovascular protective effect [48,49]. A systematic search in several databases for prospective studies investigating the associations between 12 major food groups and the risk of CRC and T2D came to the conclusion of a significant inverse association with whole grains and a positive association with red meat and processed meat in a dose-response manner [50,51], which was in line with our observation of the significant reduced CRC risk and T2D risk when exchanging plant protein for animal protein in the few studies that have been done to date. Two recent meta-analyses of randomized controlled trials summarized the substitutional effect of plant protein for animal protein on glycemic control and blood lipids in an interventional setting [52,53]. Results from 13 trials focusing on replacing animal with plant protein at a median level of ~35% of total protein per day on glycemic control in diabetes showed such a substitution led to a significantly lowered HbA1c, fasting glucose and fasting insulin [53]. The other review with 112 trials on plant protein in substitution for animal protein of ≥3 weeks found significantly decreased low-density lipoprotein cholesterol, non-high-density lipoprotein cholesterol and apolipoprotein B [52].

The underlying mechanisms for the observed protective effects of substitution of plant protein for animal protein on all-cause and CVD mortality risk as well as CRC and T2D risk may be attributable to differences between their food sources’ co-occurring bioactive nutrients or compounds, amino acid composition, and certain microbiome-generated circulating metabolites [30,51,54,55,56]. Biochemical interactions between various nutrients or compounds in the protein food source can exert health effects, such that the effects of protein intake on disease may be due to other components of the food, not necessarily the protein content. Higher red and/or processed meat consumption has been consistently linked with multiple CVD risk factors such as abnormal serum lipid level, hypertension, and insulin resistance, possibly due to the nutrient profile associated with meat intake such as dietary cholesterol, heme iron, nitrates and nitrite, and advanced glycation end-products, some of which are carcinogenic compounds also related to increased risk of CRC and premature death [12,15,54,56,57,58]. High contents of phenolic compounds, vitamins, minerals, fiber, and phytoestrogens in grains and cereal foods has been associated with reduced cancer and CVD risk, mainly owing to their anti-inflammatory and antioxidant effect to improve metabolic profiles [54,59,60]. In terms of amino acid composition, plant-based protein is lower in essential amino acids (e.g., methionine, lysine, and tryptophan) but contains higher nonessential amino acids (e.g., arginine and glycine) that could favorably influence cardiovascular health by decreasing blood pressure and vascular reactive oxygen species [54,61,62]. In addition, fermentation by gut bacteria of undigested protein mainly contributed by animal protein into by-products such as ammonia, phenolic and indolic compounds, and hydrogen sulfide, was reported to promote alterations in colonic epithelial cells, leading to development of CRC [63]. Similarly, through gut microbiome metabolism, L-carnitine and choline from red meat intake formed the circulating trimethylamine-N-oxide (TMAO), a metabolite that increases major adverse cardiac events and CVD mortality [64]. However, limited evidence supports the benefits of plant- over animal-based protein on maintaining muscle mass [65]. In line with our finding of substituting plant protein for animal protein being associated with reduced dementia mortality and DAI, accumulating epidemiological evidence suggests a protective role in adherence to the Mediterranean diet emphasizing more plant-based food intake in neurodegenerative diseases and brain health, which could result from plant polyphenols’ role in activating similar molecular pathways as caloric restriction diets [66].

Although there was not a large amount of evidence accumulated, current evidence on reduced risk of all-cause and CVD mortality in association with substitution of plant protein for animal protein was consistent across various food sources of animal proteins and across different diseases, and was also supported by recent reviews of the independent effects of specific protein intake and studies of biologic mechanisms. Moreover, given that the ideal isocaloric substitution of protein from different sources in an individual’s daily life was hard to achieve, the current review provided accurate scientific evidence in this aspect by deriving the isoenergetic substitutional effects of animal and plant protein based on statistical modeling. However, several limitations to the previous literature should be noted. First of all, although several included studies described an isocaloric substitution model in the analyses, they did not adjust for carbohydrate but instead adjusted for vegetables, fruits, fiber, or glycemic index in the fully adjusted model, which could potentially lead to biased results and incomparability with other studies. Secondly, even among studies with the same outcome and same protein source for substitution, different exposure data type (i.e., continuous or quintiles) or units (3% or 5% of energy) caused incomparability of study findings. Other differential aspects of included studies in this review included different study designs, different study population characteristics and disease statuses (diseased vs normal population), and results reported among different sexes (female only, male only, or both sexes combined). Thirdly, all the 9 studies using “percentage of energy” as protein substitution unit applied 1–5% of energy as exposure, but this number was arbitrarily set; since a normal person usually consume 15% to 35% of energy from protein per day, future studies may consider using a higher cut-off for analysis to be more applicable to real-life situations. The majority of previous studies were conducted among primarily European descent, limiting the generalizability of the findings. Moreover, multiple biases might have occurred in the original studies. Several important confounders such as socioeconomic status, physical activity, and medical history were not controlled in some of the included studies. One-time diet assessment in most studies might lead to measurement bias, given diet may change over time. Use of self-reported FFQs, food record or other questionnaires collecting information might have led to information bias and thus caused non-differential misclassification. Residual or unmeasured confounding cannot be completely ruled out in observational studies. Our review is also limited by the scope of isocaloric substitution of two protein types; exchanging protein for fat or carbohydrate under isoenergetic condition was not investigated, though they are also common dietary substitution situations. Finally, there were a limited number of studies on each particular disease outcome, especially on cancers and metabolic diseases. In addition, no studies have focused on joint conditions including arthritis or other rheumatic disorders, one of the most prevailing aging-related outcomes in most Western countries. Future studies on these disease outcomes are warranted to provide a more conclusive summarization.

## 5. Conclusions

In summary, findings from this review suggested isocaloric substitution of plant-based protein for animal-based protein was inversely associated with risk of all-cause and CVD mortality, with the protective effect primarily contributed by substituting bread, cereal, and pasta protein for red meat protein. Given the limited number of studies on each outcome of this review, more studies with different aging-related health outcomes and diverse study populations are needed to accumulate more evidence and confirm our findings. These preliminary findings may provide important public health implications as well as recommendations of introducing plant protein-rich sources to replace animal proteins to prevent aging-related diseases, and promote longevity and healthy aging.

## Figures and Tables

**Figure 1 nutrients-14-00272-f001:**
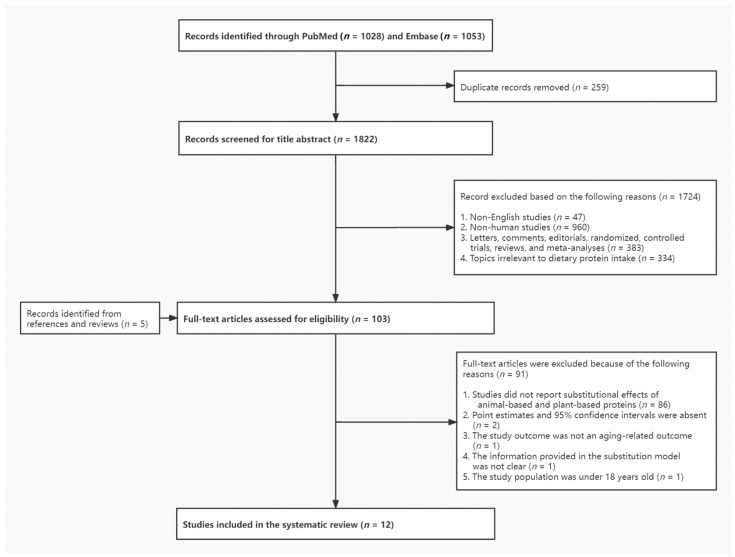
Flow diagram of literature search process.

**Figure 2 nutrients-14-00272-f002:**
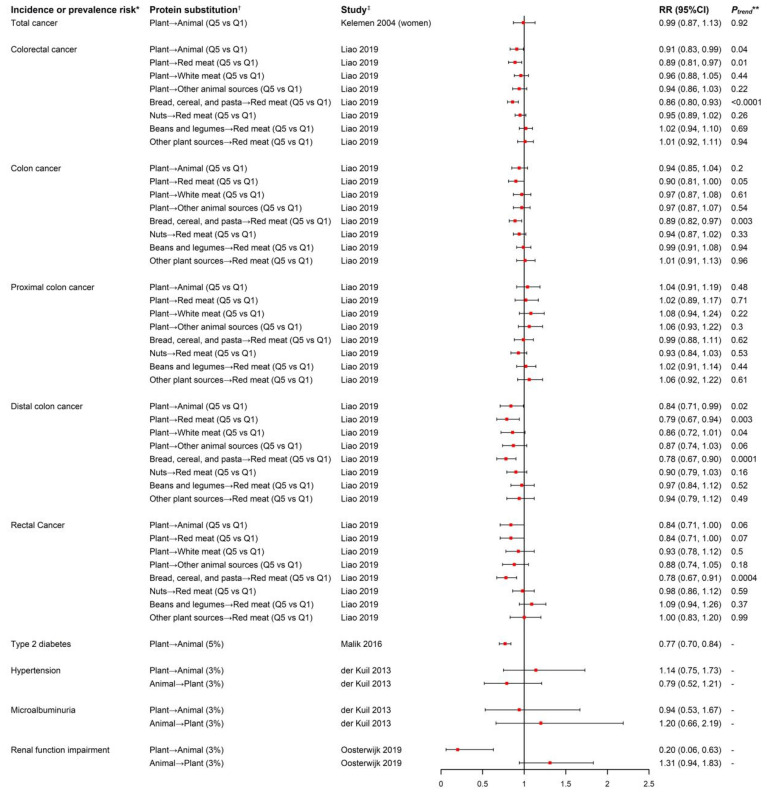
Forest plot of incidence or prevalence risk of aging-related diseases in relation to isocaloric substitution of plant-based protein and animal-based protein. * In the study by Oosterwijk et al., the outcome was prevalence risk of renal function impairment while the outcome was incidence risk in the rest of the studies. ^†^ Symbol “→” represents substitution of protein from the left-side food source for protein from the right-side food source. “Plant” and “Animal” stands for “all the plant-based food sources” and “all the animal-based food sources”, respectively. The content in the bracket after the substitution exposure describes the substituted exposure’s unit and data type: “Q5 vs. Q1” in the study by Kelemen et al. was substitution of Quintile 5 versus Quintile 1 of percentage of energy from total plant protein for total animal protein. In the study by Liao et al., the exposure was the substitution of Quintile 5 versus Quintile 1 of amount of plant-based protein for animal-based protein. “3%” or “5%” in the rest of studies referred to substitution of 3% or 5% of total energy intake from various plant-based proteins for various animal-based proteins. ^‡^ The content in the bracket after reference of publication indicated the population in the analysis, both sexes combined if not otherwise indicated. ** The *P*_trend_ value was only reported for the categorical substituted exposure.

**Figure 3 nutrients-14-00272-f003:**
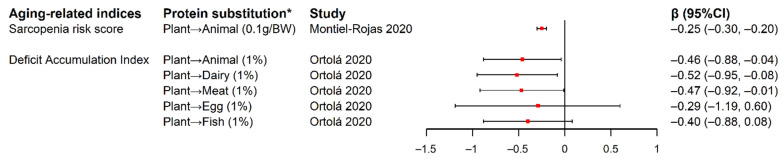
Forest plot of estimated changes and 95% CIs in aging-related indices in relation to isocaloric substitution of plant-based protein for animal-based protein. * Symbol “→” represents substitution of protein from the left-side food source for protein from the right-side food source. “Plant” and “Animal” stands for “all the plant-based food sources” and “all the animal-based food sources”, respectively. The content in the bracket after the substitution exposure describes the substituted exposure’s unit and data type: “0.1g/BW” in the study by Montiel-Rojas et al. was substitution of 0.1g total plant protein per body weight for the detriment of total animal protein. “1%” in the study by Ortolá et al. referred to substitution of 1% of total energy intake from total plant protein for various animal-based proteins. Abbreviation: BW, body weight; CI, confidence interval.

**Table 1 nutrients-14-00272-t001:** Population, Intervention, Comparison, Outcomes, Study (PICOS) criteria for inclusion of studies.

Parameter	Inclusion Criteria
Population	Adults aged over 18 (including mean age) at baseline for cohort studies
Intervention/exposure	Isocaloric substitution of plant-based and animal-based protein or protein from food sources, as defined in the context of nutritional substitution model
Comparison	Continuous (e.g., 3% or 5% of total energy from plant protein substituted for animal protein) or categorical (e.g., highest vs. lowest level of percentage of energy from plant protein substituted for animal protein)
Outcomes	Aging-related health outcomes which included mortality outcomes, aging-related disorders such as cancer, type-2 diabetes, chronic kidney diseases, cardiometabolic diseases and risk markers, as well as aging-related indices
Study design	Original research studies of any observational design were eligible. Systematic or narrative reviews, intervention studies, conference or dissertations, editorials, case reports or other descriptive studies were excluded

**Table 2 nutrients-14-00272-t002:** Summary characteristics of included studies on the association between isocaloric substitutional effect of animal-based and plant-based protein and aging-related health outcomes (*N* = 12).

Reference	Study Design (Location)	Number of Cases ^a^/Total Individuals at Risk ^b^	Mean Age (Range)	Mean or Median Follow-Up Time for Cohort Study	Diet Assessment Instrument/Assessment Period/Whether Assessment of Validity and Reproducibility	Substitutional Model	Aging-Related Outcomes	Variables for Adjustment ^c^
Kelemen et al. (2004) [31]	Cohort study (USA)	4843 total incident cancer cases and 3978 total deaths/29,017 participants	55–69	Mean = 11.4 years	131-item self-administered FFQ/in the past year/validity and reproducibility were both assessed	Highest versus lowest quintile of percentage of energy from total plant protein substituted for total animal protein while holding constant the intake of total energy, carbohydrate and fat	1. Mortality from the following causes: all-cause, CHD, total cancer; 2. Total cancer incidence	Age, total energy, carbohydrate, saturated fat, polyunsaturated fat, monounsaturated fat, trans-fat, total fiber, dietary cholesterol, dietary methionine, alcohol drink, smoking status, activity level, BMI, history of hypertension, postmenopausal hormone use, multivitamin use, vitamin E supplement use, education, and family history of cancer
der Kuil et al. (2013) [37]	Cohort study (16 European countries)	298 incident hypertension cases/1319 participants with type-1 DM	31.0 (15–60)	Mean = 7 years	3-day food record/within a 2-week period at baseline/NA	1.Substitution of 3% of energy intake form total animal protein for total plant protein holding constant the intake of total energy, carbohydrate and fat; 2. Substitution of 3% of energy intake from total plant protein for total animal protein holding constant the intake of total energy, carbohydrate and fat	Hypertension and microalbuminuria incidence	Age, sex, diabetes duration, HbA1c, BMI, smoking status, physical activity, total energy intake, energy densities from fat, carbohydrate and alcohol
		135 incident microalbuminuria cases/1045 participants with type-1 DM						
Malik et al. (2016) [33]	Cohort studies (USA) ^d^	7214 incident type-2 DM cases/72,992 participants	30–55	Mean = 20.2 years	131-item self-administered FFQ/in the past year/validity and reproducibility were both assessed	Substitution of 5% of energy intake from total plant protein for total animal protein holding constant the intake of total energy and fat	Type-2 diabetes incidence	Age, family history of diabetes, smoking status, alcohol intake, physical activity, race/ethnicity, postmenopausal hormone use, oral contraceptive use, total energy intake, percentage of energy from fat, dietary cholesterol, dietary fiber, glycemic index, and BMI
		5032 incident type-2 DM cases/92,088 participants	24–42					
		3334 incident type-2 DM cases/40,722 participants	40–75					
Song et al. (2016) [10]	Cohort study (USA)	36,115 total deaths/131,342 participants	49 (30–75)	Mean = 27.0 years	131-item self-administered FFQ/in the past year/validity and reproducibility were both assessed	Substitution of 3% of energy intake from total plant protein for animal proteins from various animal-based food sources (i.e., processed red meat, unprocessed red meat, poultry, fish, egg, dairy) holding constant the intake of total energy, and fat	Mortality from the following causes: all-cause, CVD, total cancer	Total caloric intake, age, sex, percentage of energy from saturated fat, polyunsaturated fat, monounsaturated fat, trans-fat, multivitamin use, smoking status, pack-years of smoking, BMI, physical activity, alcohol consumption, history of hypertension diagnosis, glycemic index, and intake of whole grains, total fiber, fruits and vegetables.
Van Baak et al. (2017) [36]	Cross-sectional study (8 European countries)	489 overweight or obese participants	42.3 (<65)	NA	3-day food record/Four weeks after the start of WM phase and in the last week of WM phase ^e^/NA	Substitution of 1% of total protein intake from total animal protein for total plant protein holding constant the intake of total protein	Change in body weight, body fat, waist circumference, SBP, DBP, total cholesterol, HDL-C, LDL-C, triglycerides, fasting glucose, fasting insulin, HOMA-IR, matsuda index, CRP, adiponectin during the WM phase	BMI at randomization, changes in the anthropometrics, blood pressure and metabolic parameters during the weight loss phase, gender, type of center, dietary protein intake, glycemic index, dietary fat intake and fiber intake
Budhathoki et al. (2019) [29]	Cohort study (Japan)	12,381 total deaths/70,696 participants	55.7 (45–74)	Mean = 18 years	138-item self-administered FFQ/in the past year/validity and reproducibility were both assessed	Substitution of 3% of energy intake from total plant protein for animal proteins from various animal-based food sources (i.e., red meat, processed meat, chicken, egg, dairy, fish) holding constant the intake of total energy, carbohydrate and fat	Mortality from the following causes: all-cause, CVD, total cancer	Total energy, percentage of energy from fats and carbohydrates, age, sex, BMI, smoking status, alcohol use, physical activity, occupation status, and intake of green tea and coffee.
Liao et al. (2019) [32]	Cohort study (USA)	8995 incident colorectal cancer cases/489,625 participants	50–71	Median = 15.5 years	124-item self-administered FFQ/in the past year/validity and reproducibility were both assessed	1. Highest versus lowest quintile of amount of total plant protein substituted for animal protein from various animal-based food sources (all animal foods, red meat, white meat, other animal foods) holding constant the intake of total energy and protein; 2. Highest versus lowest quintile of amount of plant protein from various plant-based food sources (bread, cereal and pasta; nuts; beans and legumes; other plant sources) substituted for red meat protein holding constant the intake of total energy and protein	Colorectal cancer, colon cancer, proximal colon cancer, distal colon cancer, and rectal cancer incidence	Age, total protein, total energy, sex, education, marriage status, family history of colon cancer, race, BMI, smoking status, frequency of vigorous physical activity, alcohol intake, fruit intake, vegetable intake, total calcium intake, total folate intake, dietary fiber intake.
Oosterwijk et al. (2019) [38]	Cross-sectional study (Netherland)	99 renal function impairment cases/420 participants with type-2 DM	63	NA	177-item self-administered FFQ/in the past month/only validity was assessed	1. Substitution of 3% of energy intake from total plant protein for total animal protein holding constant the intake of total energy, fat and carbohydrate; 2. Substitution of 3% of energy intake from total animal protein for total plant protein holding constant the intake of total energy, fat and carbohydrate.	Renal function impairment prevalence	Age, gender, diabetes duration, BMI, smoking status, physical activity, alcohol intake, saturated fat intake, unsaturated fat intake, intake of mono- and disaccharides, intake of polysaccharides, intake of fiber and intake of trans fatty acids.
Huang et al. (2020) [30]	Cohort study (USA)	77,614 total deaths/416,104 participants	62.1 (50–71)	Median = 15.5 years	124-item self-administered FFQ/in the past year/validity and reproducibility were both assessed	1. Substitution of 3% of energy from total plant protein for animal proteins from various animal-based food sources (all animal foods, red meat, white meat, dairy, egg) holding constant the intake of total energy and fat; 2. Substitution of 3% of energy from plant protein from various plant-based food sources (bread, cereal and pasta; nuts; beans and legumes; other plant foods) for egg and red meat protein holding constant the intake of total energy and fat.	Mortality from the following causes: all-cause, CVD, total cancer, heart disease, stroke, respiratory disease	Age at entry, BMI, alcohol consumption, smoking status, physical activity, race or ethnic group, educational level, marital status, diabetes, health status, vitamin supplement use, daily dietary total energy, animal protein, saturated fat, polyunsaturated fat, monounsaturated fat, trans fat, fiber, vegetables, and fruits, and postmenopausal hormone replacement therapy.
Ortolá et al. (2020) [7]	Cohort study (Spain)	812 participants	68.6 (>60)	Median = 8.2 years	900-item computerized diet history/in the past year/ only validity was assessed	1% change in energy from total plant protein substituted for animal protein from various animal-based sources (total animal foods, dairy, meat, egg and fish) from wave 0 (2008–2010) to wave 1 (2012) holding constant the intake of total energy, carbohydrate and fat	Change in DAI between wave 0 (2008–2010) and wave 3 (2017) ^f^	Sex, age, educational level, DAI at wave 0, changes in energy intake, vegetable protein intake, animal protein intake from all sources except the one being examined, fat intake, carbohydrate intake and alcohol intake from wave 0 to wave 1, changes in smoking status, alcohol consumption status, leisure-time physical activity, sedentary behavior, and BMI from wave 0 to wave 3.
Montiel-Rojas et al. (2020) [34]	Cross-sectional study (4 European countries)	986 participants	65–79	NA	1-week food record/at baseline/ NA	Substitution of 0.1g/BW increase of total plant protein for total animal protein holding constant the intake of total energy and protein	Sarcopenia risk score ^g^	Total protein intake, plant protein intake, total energy intake, age, recruiting center, medication, smoking habits, prevalence of MetS, adherence to PA guidelines, and fiber intake.
Sun et al. (2021) [35]	Cohort study (USA)	25,976 total deaths/102,521 participants	50–79	Mean = 18.1 years	122-item self-administered FFQ/in the past three months/validity and reproducibility were both assessed	Substitution of 5% of energy from total plant protein for total animal protein holding constant the intake of total energy and fat	Mortality from the following causes: all-cause, CVD, total cancer, dementia	Age at baseline, race/ethnicity, education, income, Observational Study/Clinical Trials, hormone use history, smoking status, physical activity, baseline diabetes mellitus status and high blood cholesterol status, and family history of heart attack/stroke, alcohol intake, total energy intake, percentage of energy from saturated fatty acids, polyunsaturated fatty acids, monounsaturated fatty acids and trans-fatty acids, dietary fiber intake, and glycemic load.

Abbreviations: BMI: body mass index; BW: body weight; CHD: coronary heart disease; CRP: C-reactive protein; CVD: cardiovascular diseases; DAI: deficit accumulation index; DBP: diastolic blood pressure; FFQ: food frequency questionnaire; HDL-C: high-density lipoprotein cholesterol; HOMA-IR: Homeostatic Model Assessment for Insulin Resistance; LDL-C: low-density lipoprotein cholesterol; MetS: metabolic syndrome; NA: not applicable; PA: physical activity; SBP: systolic blood pressure; WM: weight maintenance. ^a^ If a study had all-cause mortality and cause-specific morality as outcomes, we only reported the number of total deaths. For non-mortality outcomes, we reported number of cases for each disease outcome unless the outcome was continuous measurement (instead of disease status) for which we only reported all the participants in the study. ^b^ Population at risk are normal disease-free individuals unless otherwise noted. ^c^ Covariates were those adjusted in the final fully adjusted substitutional association model. ^d^ The substitutional effect estimates from three cohorts were computed by fixed-effects meta-analysis. ^e^ In this 6-month weight maintenance study following an energy-restricted diet for weight loss, dietary intake during the 26-week was calculated as the mean intake reported in the 3-day food diaries at week 4 and week 26. ^f^ The DAI was calculated as the total sum of points assigned to each deficit divided by the number of deficits considered (52 in total for 4 domains: functional impairments, self-reported health/vitality, mental health, and morbidities/use of health services) and further multiplied by 100 to obtain a range from 0 (lowest) to 100% (highest deficit accumulation). ^g^ The sarcopenia risk score was the composite z-scores calculated and averaged by the sex-specific standardized values of skeletal muscle mass index and handgrip strength.

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
