# Peer review of "The Isocaloric Substitution of Plant-Based and Animal-Based Protein in Relation to Aging-Related Health Outcomes: A Systematic Review"

_nutrients, 2022, doi:10.3390/nu14020272_

Round 1

Reviewer 1 Report

This is a well written and appropriately structured systematic review of the literature dealing with the isocaloric substitution of plant-based and animal-based protein in relation to aging-related health outcomes. 

My only concern is that figure 2 is very difficult to read as written on one page and needs to be split into smaller sections so that the fonts are at least as big as figure 3.

Reviewer 2 Report

The paper is well written and the different parts are well described, analyzed and discussed. I don't propose any minor or major revision for this paper.

Reviewer 3 Report

  • Important and well written review. 
  • Include the fact that this study represents the data of 1.45 million subjects in your summary.
  • Is this a qualitative review? Explain why this is a qualitative review, or omit 'qualitative'.
  • In table 1 you mention that reviews are excluded, but in the text it is mentioned that you included 1 meta-analysis (which refers to a study which is not a meta-analysis, ref. 33). If you truly included a meta-analysis it is better to include the original studies. 
  • Table 2 and 3 should be either better divided or summarized in one table. Some suggestions:
    •  Location and collection period less relevant (table 2). Combine design & location with just the country (e.g. cohort study, USA).
    •  Combine number of cases and total individuals at risk in 1 column.
    • Since NOS is 7-9 for all studies, I would recommend leaving that out of the table.
    • Dietary assessment instrument, period and assessment of validity/reproducibility could be combined in one column.
  • If table 2 and 3 are more or less kept this way, then table 3 could be more condensed.
  • Something went wrong with reference 33. In Table 2 you refer to Malik et al. in 2016, but the study you refer to is from 2020? It seems the same study you refer to in reference 34, but that one is missing in the reference list. Please correct this.
  • Sometimes sentences were very long (e.g. 229-235, 457-463) and lost clarity. Especially 457-463 was totally unclear to me (explain "degree and speed of aging").
  • Figure 2: perhaps this figure could be placed in an appendix, while leaving a summary in the main text. In the current figure it's mainly a list of results from Huang 2020 with quite some substitutions for which the CI is extremely large, suggesting (e.g. for legumes - egg) that too little data were available.
  • In Figure 3 the p-values for Type 2 diabetes and below are lacking.
  • If possible, I would add figure 4 for results from paragraph 3.5 (unhealthy aging). 
  • The difference in the "gender effect of soy and isoflavones intake on CVD mortality that was supported in previous cohort studies" (sentence 467-469) is very interesting and should be explained some more.
  • Joint conditions are now in most Western countries among the most prevailing chronic conditions. This should be mentioned in the discussion, as there were apparently no studies including arthritis or other rheumatic disorders.
  • Most studies were based on a 1-5% substitution of protein. Isn't that too low? Why not 10%? Perhaps the numbers would be surprisingly larger (i.e. favoring plant-based proteins much more). I do think that the 1-5% substitution should be clearly stated as a limitation, since it is very arbitrary. (Average animal protein intake is 60% in western countries, so you would go tot 55-59% with these levels of substitution). 
  • For the discussion: please explain more clearly why substitution of the same percentage of animal protein by plant protein vs. the substitution of plant by animal protein does not lead to exactly the opposite relative risk? (van der Kuil 2013, and Oosterwijk, 2019, in Figure 3).
  • For the discussion: please include evidence from intervention studies.
  • Reference numbers in text: please add a space before opening parentheses.
